# Assessment of four *in vitro* phenotypic biofilm detection methods in relation to antimicrobial resistance in aerobic clinical bacterial isolates

Ajaya Basnet[1,2], Basanta Tamang[2]*, Mahendra Raj Shrestha[3], Lok Bahadur Shrestha[4], Junu Richhinbung Rai[5], Rajendra Maharjan[3], Sushila Dahal[6], Pradip Shrestha[6], Shiba Kumar Rai[7]

1 Department of Medical Microbiology, Shi-Gan International College of Science and Technology, Maharajgunj, Kathmandu, Nepal, 2 Department of Microbiology, Nepal Armed Police Force Hospital, Balambu, Kathmandu, Nepal, 3 Department of Clinical Laboratory, Nepal Armed Police Force Hospital, Balambu, Kathmandu, Nepal, 4 School of Medical Sciences and The Kirby Institute, University of New South Wales, Sydney, Australia, 5 Department of Microbiology, Maharajgunj Medical Campus, Tribhuvan University Teaching Hospital, Institute of Medicine, Maharajgunj, Kathmandu, Nepal, 6 Department of Microbiology, KIST Medical College and Teaching Hospital, Balambu, Kathmandu, Nepal, 7 Department of Research and Microbiology, Nepal Medical College and Teaching Hospital, Attarkhel, Kathmandu, Nepal

* tamangbasanta222@gmail.com

**Data Availability Statement:** The data of the manuscript has been deposited in a public repository. URL: https://datadryad.org/stash/

## Abstract

### Introduction

The lack of standardized methods for detecting biofilms continues to pose a challenge to microbiological diagnostics since biofilm-mediated infections induce persistent and recurrent infections in humans that often defy treatment with common antibiotics. This study aimed to evaluate diagnostic parameters of four *in vitro* phenotypic biofilm detection assays in relation to antimicrobial resistance in aerobic clinical bacterial isolates.

### Methods

In this cross-sectional study, bacterial strains from clinical samples were isolated and identified following the standard microbiological guidelines. The antibiotic resistance profile was assessed through the Kirby-Bauer disc diffusion method. Biofilm formation was detected by gold standard tissue culture plate method (TCPM), tube method (TM), Congo red agar (CRA), and modified Congo red agar (MCRA). Statistical analyses were performed using SPSS version 17.0, with a significant association considered at p<0.05.

### Result

Among the total isolates (n = 226), TCPM detected 140 (61.95%) biofilm producers, with CoNS (9/9) (p<0.001) as the predominant biofilm former. When compared to TCPM, TM (n = 119) (p<0.001) showed 90.8% sensitivity and 70.1% specificity, CRA (n = 88) (p = 0.123) showed 68.2% sensitivity and 42% specificity, and MCRA (n = 86) (p = 0.442) showed 65.1% sensitivity and 40% specificity. Juxtaposed to CRA, colonies formed on MCRA developed more intense black pigmentation from 24 to 96 hours. There were 77 multi-drug-

dataset/doi:10.5061/dryad.wh70rxwtg DOI: https://doi.org/10.5061/dryad.wh70rxwtg.

**Funding:** The author(s) received no specific funding for this work.

**Competing interests:** The authors have declared that no competing interests exist.

resistant (MDR)-biofilm formers and 39 extensively drug-resistant (XDR)-biofilm formers, with 100% resistance to ampicillin and ceftazidime, respectively.

## Conclusion

It is suggested that TM be used for biofilm detection, after TCPM. Unlike MCRA, black pigmentation in colonies formed on CRA declined with time. MDR- and XDR-biofilm formers were frequent among the clinical isolates.

## Introduction

Bacterial biofilms are an organized community of cells that are irreversibly attached to a surface and enveloped in a complex of self-produced extracellular polymeric substances (EPS) [1]. It is in the host's defense against hostile immune responses like opsonization and phagocytosis that bacteria form biofilms; however, during this growth mode, they can cause chronic tissue infections, as well as device-related infections like those on orthopedic alloplastic devices, endotracheal tubes, and catheters [2,3].

Once established, biofilm-mediated infections become very difficult to eradicate with antimicrobials [4], since antimicrobial resistance in biofilms can be up to 1,000 times greater than in planktonic forms and is often conferred by multi-factorial resistance mechanisms [5]. For instance, the glycocalyx layer of the biofilm accumulates antibacterial molecules up to 25% of its weight and limits the transportation of antibiotics to the bacterial cells embedded in the community [5]. Similarly, studies have shown metabolically inactive, non-dividing persisters —the survivors from antimicrobial activity—may account for re-inducing the growth of bacterial biofilms on the termination of antibiotic treatment, often with reduced susceptibility [2,5]. Occasionally, bacteria may even enter a genetic dormancy state under stressful environmental conditions, resulting in the expression of multiple drug resistance (MDR) pumps and alterations in the profiles of outer membrane proteins [5,6]. In addition, different bacterial species may contribute to the development of biofilms, which may open up the possibility of quorum sensing for horizontal gene transmission within the spatially well-organized architecture [4].

As biofilm-mediated infections pose a serious threat to hospitalized patients due to nosocomial diseases, often associated with treatment failure, their detection is vital for prevention [7]. Different genotypic and phenotypic methods have come into the light for the detection of biofilms. Electron microscopy and molecular methods (polymerase chain reaction and sonication) can be used to examine biofilms for a more sensitive and specific analysis; however, the increased operational costs and the requirement for qualified human resources for their operation limit their application, mandating the need for an alternative [8]. Phenotypic methods, such as the tissue culture plate method (TCPM) (quantitative), tube method (TM) (qualitative), Congo red agar (CRA) (qualitative), and modified Congo red agar (MCRA) (qualitative), can be used to detect biofilm-forming properties in a simple, rapid, and relatively sensitive manner [9–11]. Despite the abundance of literature on biofilm formation, only a few studies have evaluated the biofilm detection accuracy of the former three methods, and even fewer on MCRA.

In developing countries like Nepal, research on biofilms is rare; however, in light of the burden of biofilm-associated infections and the lack of standardized phenotypic methods for detecting biofilm formation, this study attempted to evaluate diagnostic parameters of four *in vitro* phenotypic methods that can be used to detect biofilms routinely.

## Materials and methods

### Study design

A descriptive cross-sectional study was conducted in the Department of Clinical Microbiology, Nepal Armed Police Force Hospital (NAPFH), Kathmandu, Nepal, between September 2021 and February 2022.

The Department of Clinical Laboratory, Nepal Armed Police Force Hospital initially provided a no objection certificate (Reference No.: 1335) with explicit permission to conduct the study. Following a proposal defense, ethical approval (Reference No.: 20790103) for the study was obtained from the Institutional Review Committee of the Shi-Gan Health Foundation, Kathmandu, Nepal. Following the ethical approval, the study protocol was not altered in any way. This study was rigorously based on the specimens of hospital-visiting patients who were suspected of infection. These samples were processed for routine hospital analysis, and not just for scientific study. Therefore, only study participants (adult participant, or parent or legal guardian in the case of minors) who provided written informed consent were included in the study.

### Sample collection

The microbiology laboratory received a total of 1,012 clinical samples, including blood, central venous catheter (CVC) tips, urine, sputum, ascitic and pleural fluids, and pus and wound swabs, from the hospital's outpatient and inpatient departments for the microbial culture and sensitivity testing.

### Isolation and identification of clinical isolates

Aseptically drawn blood samples were inoculated into a biphasic brain-heart infusion medium. Except for urine samples that were streaked on cysteine lactose electrolyte deficient (CLED) agar, all samples were streaked on blood agar, chocolate agar, and MacConkey agar. The streaked culture plates were incubated aerobically for 24 hours at 37°C. If the colonies formed within the first 24 hours, they were examined macroscopically before being subjected to microbiological tests, such as Gram staining and biochemical tests, to identify the pathogens. Before declaring the results as sterile, body fluids and CVC tip cultures were re-incubated for an additional 24 to 48 hours, and blood culture samples for an additional 48 to 72 hours. The collected samples were processed under Standard Microbiological Procedure for the isolation, identification, and characterization of organisms using the Manual of Clinical Microbiology [12].

### Antimicrobial susceptibility testing

The identified isolates were tested for antimicrobial susceptibility testing using Kirby Bauer's disc diffusion method on Mueller Hinton agar, following the most recent 30[th] guideline of the Clinical Laboratory Standards Institute (CLSI) [13].

The bacterial isolates were stored in a microcentrifuge tube containing tryptic soy broth (TSB) with 30% glycerol at -70°C until further investigation.

### Definition of types of antimicrobial resistances

MDR was defined as acquired non-susceptibility to at least one agent in three or more antimicrobial categories; extensively drug resistance (XDR) was defined as non-susceptibility to at least one agent in all but two or fewer antimicrobial categories (i.e., bacterial isolates remain susceptible to only one or two categories) [5].

## Detection of biofilm formation

**Tissue Culture Plate Method (TCPM).** The TCPM for biofilm detection was performed as per the guidelines of Christensen et al. [14], which is considered the gold-standard method for biofilm detection. A single bacterial colony isolated from a fresh agar plate was emulsified in normal saline by standardizing with 0.5 McFarland turbidity standards (i.e., $1.5 \times 10^8$ CFU/ml). The bacterial suspension was diluted (1:100) in a fresh tryptic soy broth medium, and 200 μL of this solution was inoculated onto the sterile flat-bottomed 96-well polystyrene microtiter plate. The inoculated plate was incubated for 24 hours at 37°C. Following incubation, the wells were washed three times with phosphate-buffered saline (pH 7.2) and gently tapped to remove free-floating bacteria and well contents. The biofilm formed by bacteria adhered to the wells was dried in an inverted position at room temperature, which was followed by fixation with 2% sodium acetate. After that, the plate was then stained with 0.1% crystal violet (CV) solution for about 10–15 minutes. Again, the plate was shaken vigorously and washed three times with phosphate buffer saline. At this point, bacterial adhesions in the wells could be seen macroscopically. Finally, the bounded CV dye was resolubilized in 30% acetic acid for 30 minutes and measured in an ELISA reader at 570 nm to determine the optical density (OD).

The cut-off OD (ODc) was defined as equivalent to three standard deviations above the mean OD of the negative control (sterile broth). Three categories of isolates were identified (Table 1). The assays were performed in triplicate.

**Tube method (TM).** TM method for biofilm detection was performed as per the guidelines of Christensen et al [15]. Test isolates after being grown on an agar plate for 24 hours were inoculated into polystyrene tubes containing tryptic soy broth with 1% glucose. A 0.5 McFarland standard was used to adjust the turbidity. The inoculated tubes were incubated overnight at 37°C. Each tube's contents were cautiously aspirated with a pipette before the tubes were washed three times with phosphate buffer saline (pH 7.2). The tubes were stained for 15 minutes with 0.1% CV solution, decanted, and then washed similarly. Tubes were dried in an inverted position at room temperature and were macroscopically examined for the development of biofilm. Positive tubes had stained films or adherent layers on the interior of the tube. All assays were performed in triplicate.

**Congo Red Agar method (CRA).** CRA for biofilm detection was performed as per the guidelines of Freeman et al [16]. CRA medium was prepared with brain heart infusion broth 37 g/L, sucrose 50 g/L, agar (No. 1) 10g/L, and Congo red indicator 0.8 g/L. Congo red stain was prepared as a concentrated aqueous solution and autoclaved (121°C for 15 minutes) separately from the other medium components. Once the temperature reached 55°C, the dye (autoclaved) was then added to the brain-heart infusion broth (autoclaved) that contained agar and sucrose. Test organisms were inoculated onto CRA plates, which were then incubated aerobically at 37°C for 24 hours. In contrast to red colonies, which were interpreted as being produced by strains that did not produce biofilm, black colonies with a dry crystalline consistency were indicated as strains that produce biofilm. The assays were performed in triplicate.

**Table 1. Grading of biofilm formation.**

|  | Optical densities | Rule | Biofilm formation |
|---|---|---|---|
| a) | < 0.494 | $OD_{test} < OD_c$ | None/weak |
| b) | 0.494–0.986 | $OD_c < OD_{test} < 2 \times OD_c$ | Moderate |
| c) | > 0.986 | $2 \times OD_c < OD_{test} < 4 \times OD_c$ | Strong |

**Modified Congo Red Agar method (MCRA).** MCRA method for biofilm detection was performed as per the guidelines of Mariana et al [17]. MCRA was modified from the original CRA by reducing Congo red indicator concentration from 0.8 g/L to 0.4 g/L, substituting glucose for sucrose 10 gm/L, and replacing brain-heart infusion broth with blood base agar 40 g/L. The inoculated agar was incubated for 48 hours at 37°C and then for an additional 2 to 4 days at room temperature. Indications of biofilm production were black colonies with a dry, crystalline consistency. The assays were performed in triplicate.

## Statistical analysis

Statistical Package for the Social Sciences software version 17.0 was used to perform calculations and statistical analyses. The experimental data were shown in numbers (*n*) and percentages (%). The isolated bacteria were divided into three main groups: Enterobacterales, non-fermenters, and gram-positive cocci (GPC). Based on these categories, the statistical significance of variables, such as antibiotic resistance, biofilm formation, and others, was calculated. Each *in vitro* phenotypic method's sensitivity, specificity, positive predictive value (PPV), and negative predictive value (NPV) were calculated considering TCPM as the gold standard method for biofilm detection. The Chi-square test was used to determine the statistical significance between the variables. A p-value <0.05 was considered statistically significant.

## Results

### Detection of bacterial isolates from different clinical samples

Among the total clinical samples (n = 1,012), 194 (19.17%) were culture-positive. There were 101 (52.06%) urine samples, 71 (36%) sputum samples, and 33 (17.01%) blood samples. While 162 (83.51%) samples yielded monobacterial growth, 32 (16.49%) samples yielded polybacterial growth. *Citrobacter* spp. (39.04%, 57/146), *Acinetobacter calcoaceticus-baumanii* (ACB) complex (61.9%, 26/42), and *S. aureus* (71.05%, 27/38), were predominant among Enterobacterales (146/226), non-fermenters (42/226), and gram-positive cocci (38/226), respectively. *E. coli* (78.85%, 41/52) (p<0.001) and CoNS (88.89%, 8/9) (p = 0.006) were frequently isolated in urine samples. Significant growth (p<0.05) of non-fermenters was observed in sputum samples (54.67%, 23/42) (Table 2).

### Antimicrobial resistance among the isolates

The isolates exhibited a variable level of resistance to several tested antibiotics (Table 3). Eighty-three percent (390/469) of all isolates tested were resistant to cephalosporins, 56.11% (147/262) to penicillin, and 44.64% (200/448) to aminoglycosides. A resistance rate exceeding 60% was observed for cotrimoxazole (132/208), erythromycin (23/38), and clindamycin (24/36). Among the Enterobacterales, *Klebsiella* spp. (58.33%, 14/24) (p = 0.021), followed by and *E. coli* (38.46%, 20/52) (p = 0.89) exhibited highest resistance to amoxycillin clavulanate. *Proteus* spp. (6/6) had 100% resistance to ciprofloxacin (p = 0.1), ofloxacin (p = 0.042), and cotrimoxazole (p = 0.063). *S.* Typhi had a resistance of 14.29% to cotrimoxazole (p = 0.005), gentamicin (p = 0.106), and amikacin (p = 0.068). However, *Citrobacter* spp. exhibited higher resistance to gentamicin (59.65%, 34/57) (p = 0.002) and amikacin (63.16%, 36/52) (p = 0.003). In non-fermenters, the ACB complex showed high resistance to ceftazidime (96.15%, 25/26) (p<0.001), ciprofloxacin (76.92%, 20/26) (p = 0.159), and amikacin (61.54%, 16/26) (p = 0.21). *Pseudomonas* spp. was highly susceptible to gentamicin (75%) (p = 0.039). A high proportion of *S. aureus* was resistant to ceftazidime (96.3%, 26/27) (p<0.001), ampicillin (88.89, 24/27) (p = 0.249), and cotrimoxazole (62.96%, 17/27) (p = 0.003). Fifty-nine percent of *S. aureus*

**Table 2. Sample-wise distribution of clinical isolates.**

| | | Clinical samples (n = 194) | | | | | | | | | | | | | | | |
|---|---|---|---|---|---|---|---|---|---|---|---|---|---|---|---|---|---|
| | | Urine (n = 83) | | Blood (n = 29) | | CVC tips (n = 3) | | Sputum (n = 62) | | Pleural fluid (n = 1) | | Pus (n = 7) | | Ascitic fluid (n = 5) | | Wound (n = 4) | |
| | | *n* | p-value | *n* | p-value | *n* | p-value | *n* | p-value | *n* | p-value | *n* | p-value | *n* | p-value | *n* | p-value |
| Enterobacterales | *Citrobacter* spp. (n = 57) | 22 | 0.285 | 6 | 0.314 | 2 | 0.96 | 24 | **0.044** | 1 | 0.084 | 0 | 0.094 | 2 | 0.442 | 0 | 0.241 |
| | *E. coli* (n = 52) | 41 | **<0.001** | 7 | 0.791 | 1 | 0.669 | 2 | **<0.001** | 0 | 0.584 | 0 | 0.115 | 0 | 0.216 | 1 | 0.924 |
| | *Klebsiella* spp. (n = 24) | 8 | 0.237 | 5 | 0.36 | 0 | 0.548 | 8 | 0.831 | 0 | 0.73 | 1 | 0.86 | 0 | 436 | 2 | **0.01** |
| | *S.* Typhi (n = 7) | 0 | **0.016** | 7 | **<0.001** | 0 | 0.755 | 0 | 0.069 | 0 | 0.858 | 0 | 0.607 | 0 | 0.686 | 0 | 0.718 |
| | *Proteus* spp. (n = 6) | 1 | 0.162 | 0 | 0.305 | 0 | 0.773 | 3 | 0.32 | 0 | 0.869 | 1 | 0.078 | 0 | 0.709 | 1 | **0.005** |
| Non-fermenters | ACB complex (n = 26) | 7 | 0.503 | 4 | 0.904 | 0 | 0.53 | 13 | **0.03** | 0 | 0.718 | 1 | 0.928 | 1 | 0.547 | 0 | 0.467 |
| | *Pseudomonas* spp. (n = 16) | 2 | **0.007** | 1 | 0.326 | 0 | 0.63 | 10 | **0.005** | 0 | 0.782 | 2 | **0.044** | 1 | 0.255 | 0 | 0.578 |
| Gram-positive cocci | *S. aureus* (n = 27) | 11 | 0.66 | 3 | 0.584 | 0 | 0.521 | 9 | 0.819 | 0 | 0.712 | 3 | **0.023** | 1 | 0.574 | 0 | 0.457 |
| | CoNS (n = 9) | 8 | **0.006** | 0 | 0.206 | 0 | 0.723 | 1 | 0.18 | 0 | 0.838 | 0 | 0.558 | 0 | 0.645 | 0 | 0.681 |
| | *Enterococcus* spp. (n = 2) | 1 | 0.879 | 0 | 0.557 | 0 | 0.869 | 1 | 0.57 | 0 | 0.925 | 0 | 0.786 | 0 | 831 | 0 | 0.849 |

ACB complex = *Acinetobacter calcoaceticus baumannii* complex, CoNS = coagulase negative *Staphylococcus*, CVC = central venous catheter, bold numeric = statistically significant (p<0.05).

were resistant to cefoxitin (p = 0.293). CoNS exhibited 100% (9/9) resistance to ampicillin (p = 0.315) and clindamycin (p = 0.014), while 11.11% (1/9) showed resistance to cloxacillin (p = 0.095) and cotrimoxazole (p = 0.007) (Table 3).

## Detection of biofilm formation in bacterial isolates

Among the total bacterial isolates (n = 226), TCPM detected biofilm formation in 140 (61.95%) isolates, TM in 119 (52.65%), CRA in 88 (38.94%), and MCRA in 86 (38.05%) (Fig 1). The TCPM assay identified 86 (38.05%) isolates as none/weakly adherent, 134 (59.29%) isolates as moderately adherent, and 6 (2.65%) as strongly adherent. The black pigmentation on the CRA assay that formed on the colony of biofilm producers faded after 24 hours (from 48 to 72 hours) [Fig 1C–1E], whereas the pigmentation intensified with increasing time in the MCRA assay [Fig 1F–1H].

## Diagnostic parameters of the *in vitro* phenotypic methods

Compared to the gold standard method of biofilm detection, TCPM, the tube method had sensitivity and specificity of 90.8% and 70.1% (p<0.001). Similarly, the CRA had sensitivity and specificity of 68.2% and 42% (p = 0.123), and the MCRA had sensitivity and specificity of 65.1% and 40% (p = 0.442), respectively (Table 4).

## Distribution of biofilm formers and MDR- and XDR- isolates

Among Enterobacterales (n = 146), 56.85% (83/146) were biofilm formers, of which 70.83% (17/24) and 50.88% (29/57) were *K. pneumoniae* (p = 0.13) and *Citrobacter* spp. (p = 0.244) respectively. A strain of *E. coli* was a strong biofilm producer. There were 56 (38.36%, 56/146) MDR Enterobacterales, with 39 (46.99%, 39.83) biofilm-forming MDR Enterobacterales, and 63 (43.15%, 63/146) XDR Enterobacterales with 27 (32.53%, 27/83) biofilm-forming XDR Enterobacterales. Regarding, non-fermenters (n = 42), 59.52% (25/42) were biofilm producers. ACB complex (61.5%, 16/26) (p = 0.735) was the predominant biofilm former, followed by *Pseudomonas* spp. (56.25%, 9/16) (p = 0.735). The MDR ACB complex (9/9) (p = 0.513) and XDR *Pseudomonas* spp. (2/2) (p = 0.014) were absolutely biofilm producers. Of the gram-

**Table 3. Percentage of antibiotic resistance among the isolates.**

| | | Enterobacterales (n = 146) | | | | | Non-fermenters (n = 42) | | Gram-positive cocci (n = 38) | | |
|---|---|---|---|---|---|---|---|---|---|---|---|
| | | *Citrobacter* spp. (n = 57) | *E. coli* (n = 52) | *Klebsiella* spp. (n = 24) | *S.* Typhi (n = 7) | *Proteus* spp. (n = 6) | ACB complex (n = 26) | *Pseudomonas* spp. (n = 16) | *S. aureus* (n = 27) | CoNS (n = 9) | *Enterococcus* spp. (n = 2) |
| Fluoroquinolones | Ciprofloxacin | 40 (70.18) | 34 (65.38) | 16 (66.67) | 6 (85.71) | 6 (100) | 20 (76.92) | 9 (56.25) | 15 (55.56) | 3 (33.33) | 2 (100) |
| | Ofloxacin | 34 (59.65) | 31 (59.62) | 11 (45.83) | 6 (85.71) | **6 (100)** | 19 (73.08) | 7 (43.75) | 15 (55.56) | 5 (55.56) | 2 (100) |
| Penicillins | Ampicillin | [a] | **50 (96.15)** | [a] | 6 (85.71) | [a] | [a] | [a] | 24 (88.89) | 9 (100) | 2 (100) |
| | Cloxacillin | [b] | [b] | [b] | [b] | [b] | [b] | [b] | 11 (40.74) | 1 (11.11) | 1 (50) |
| | Amoxycillin clavulanate | [a] | 20 (38.46) | 14 (58.33) | 1 (14.29) | 2 (33.33) | [a] | [a] | 4 (14.81) | 0 | **2 (100)** |
| Cephalorporins | Cefotaxime | 50 (87.72) | 42 (80.77) | 19 (79.17) | 6 (85.71) | 4 (66.67) | 24 (92.31) | 15 (93.75) | 21 (77.78) | 5 (55.56) | [c] |
| | Ceftazidime | 51 (89.47) | 43 (82.69) | 22 (91.67) | **3 (42.86)** | 5 (83.33) | 25 (96.15) | [a] | **26 (96.3)** | **5 (55.56)** | [c] |
| | Cefoxitin | [b] | [b] | [b] | [b] | [b] | [b] | [b] | 16 (59.26) | 7 (77.78) | 1 (50) |
| Aminoglycosides | Gentamicin | **34 (59.65)** | **14 (26.92)** | 11 (45.83) | 1 (14.29) | 4 (66.67) | **15 (57.69)** | **4 (25)** | 9 (33.33) | 4 (44.44) | [c] |
| | Amikacin | **36 (63.16)** | **14 (26.92)** | **17 (70.83)** | 1 (14.29) | 2 (33.33) | **16 (61.54)** | **4 (25)** | 12 (44.44) | 2 (22.22) | [c] |
| Others | Imipenem | **39 (68.42)** | **12 (23.08)** | **17 (70.83)** | 1 (14.29) | 3 (50) | **19 (73.08)** | **4 (25)** | 4 (14.81) | 0 | **2 (100)** |
| | Cotrimoxazole | 39 (68.42) | 32 (61.54) | 16 (66.67) | **1 (14.29)** | 6 (100) | 20 (76.92) | [a] | **17 (62.96)** | **1 (11.11)** | [c] |
| | Chloramphenicol | **31 (54.39)** | **14 (26.92)** | 9 (37.5) | 1 (14.29) | 4 (66.67) | [a] | [a] | 14 (51.85) | 5 (55.56) | 2 (100) |
| | Erythromycin | [b] | [b] | [b] | [b] | [b] | [b] | [b] | 14 (51.85) | 7 (77.78) | 2 (100) |
| | Clindamycin | [b] | [b] | [b] | [b] | [b] | [b] | [b] | **15 (55.56)** | **9 (100)** | [c] |

ACB complex = *Acinetobacter calcoaceticus-baumannii complex*, CoNS = *Coagulase negative Staphylococci*, -

[a] = intrinsically resistant, -

[b] = not tested, -

[c] = not effective clinically, bold numeric = statistically significant (p<0.05).

positive isolates (n = 38), *S. aureus* (n = 3) (p = 0.088) and *Enterococcus* spp. (n = 2) (p = 0.529) were the strongest biofilm producers. Among *S. aureus* (n = 27), 59.26% (16/27) were MRSA, 57.14% (12/21) were MDR-MRSA biofilm producers, and 9.52% (2/21) were XDR-MRSA biofilm producers. MDR (8/8) (p = 0.402) and XDR-CoNS (1/1) (p = 0.835) were completely biofilm formers. Similarly, 100% (2/2) of biofilm-forming *Enterococcus* spp. were XDR strains (p<0.001) (Table 5).

## Association between antibiotics resistance and biofilm production

There was a significant association (p<0.005) between resistance to ampicillin (98.31%), cotrimoxazole (55.79%), chloramphenicol (32.56%), gentamicin (30.3%), and imipenem (29.29%) with MDR-isolates, and resistance to ceftazidime (98.73%), cefotaxime (93.83%), and amikacin (71.6%) with XDR-isolates. XDR-biofilm formers showed higher resistance to cefotaxime

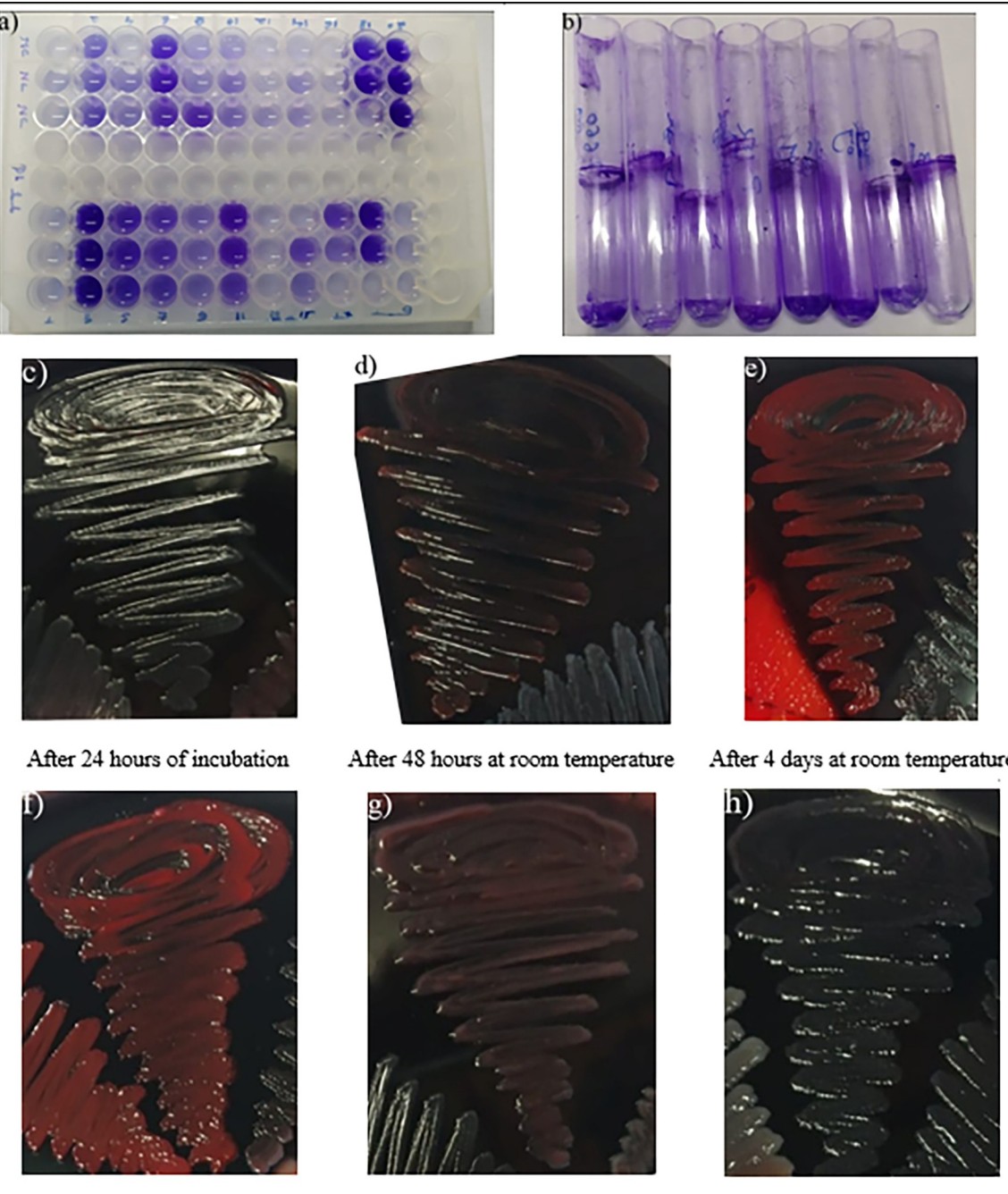

Fig. a represents biofilm formation on a tissue culture plate; Fig. b represents biofilm formation on tubes; Fig. c–e represents the fading of black pigmentation following increased hours on Congo red agar; Fig. f–h represents an acquisition of black pigmentation in the colonies with increased hours on modified Congo red agar.

**Fig 1. Phenotypic detection of bacterial biofilms.**

**Table 4. Statistical evaluation of in vitro phenotypic methods for the detection of biofilm formation.**

| | Biofilm producers | | | | p-value | Sensitivity (%) | Specificity (%) | PPV (%) | NPV (%) |
|---|---|---|---|---|---|---|---|---|---|
| | Total *n* | Non/Weak *n* (%) | Moderate *n* (%) | Strong *n* (%) | | | | | |
| TCPM | 140 | 86 (38.05) | 134 (59.29) | 6 (2.65) | - | - | - | - | - |
| TM | 119 | - | - | - | **<0.001** | 90.8 | 70.1 | 77.1 | 87.2 |
| CRA | 88 | - | - | - | 0.123 | 68.2 | 42 | 42.9 | 67.4 |
| MCRA | 86 | - | - | - | 0.442 | 65.1 | 40 | 40 | 65.1 |

TCPM = tissue culture plate method, TM = tube method, CRA = Congo red agar, MCRA = modified Congo red agar, PPV = positive predictive value, NPV = Negative Predictive Value, bold numeric = statistically significant (p<0.05).

(97.3%) (p = 0.003), cotrimoxazole (94.29%) (p<0.001), ciprofloxacin (92.31%) (p<0.001), and imipenem (84.62%) (p<0.001), while MDR-biofilm formers showed lower resistance to amoxicillin-clavulanate (18.18%) (p = 0.007), gentamicin (28.57%) (p = 0.035), and imipenem (31.17%) (p = 0.006) (Table 6).

## Discussion

Bacterial infections pose a serious public health threat and, with increasing antimicrobial resistance, a critical healthcare issue [5]. The propensity of bacteria to live in biofilm is recognized as one of the most relevant drivers of recurrent human infections and amongst the various mechanisms of antimicrobial resistance [2–4]. Despite the known medical significance, limited practices for biofilm detection in routine microbiological diagnosis are observed in low- and middle-income countries (LMICs), including Nepal. It is therefore imperative that clinicians, microbiologists, and researchers who are part of the scientific endeavor to advance better diagnosis and treatment of biofilm-associated infections be made informed of the current state of biofilm diagnosis. This study evaluated the diagnostic significance of four *in vitro* phenotypic assays for detecting biofilms and the antibiograms of biofilm formers and non-formers.

**Table 5. Incidences of MDR, XDR, and biofilm formers in the bacterial isolates.**

| | | Biofilm formers (n = 140) | p-value | MDR isolates (n = 99) | p-value | XDR isolates (n = 83) | p-value | Biofilm formers with | | | |
|---|---|---|---|---|---|---|---|---|---|---|---|
| | | | | | | | | MDR (n = 77) | p-value | XDR (n = 39) | p-value |
| Enterobacterales | *Citrobacter* spp. (n = 57) | 29 | 0.244 | 15 | **0.017** | 31 | **0.028** | 10 | 0.094 | 13 | 0.08 |
| | *E. coli* (n = 52) | 31 | 0.616 | 25 | 0.072 | 16 | **0.025** | 15 | 0.844 | 8 | 0.313 |
| | *Klebsiella* spp. (n = 24) | 17 | 0.13 | 10 | 0.715 | 11 | 0.772 | 10 | 0.273 | 5 | 0.758 |
| | *S.* Typhi (n = 7) | 4 | 0.987 | 4 | 0.295 | 1 | 0.114 | 2 | 0.902 | 1 | 0.742 |
| | *Proteus* spp. (n = 6) | 2 | 0.235 | 2 | 0.796 | 4 | 0.235 | 2 | 0.128 | 0 | 0.32 |
| Non-fermenters | ACB complex (n = 26) | 16 | 0.735 | 9 | 0.513 | 13 | **0.014** | 9 | 0.271 | 5 | 0.629 |
| | *Pseudomonas* spp. (n = 16) | 9 | 0.735 | 4 | 0.513 | 2 | **0.014** | 3 | 0.271 | 2 | 0.629 |
| Gram-positive cocci | *S. aureus* (n = 27) | 21 | 0.088 | 22 | 0.548 | 2 | 0.1 | 18 | 0.371 | 2 | 0.189 |
| | CoNS (n = 9) | 9 | 0.137 | 8 | 0.402 | 1 | 0.835 | 8 | 0.489 | 1 | 0.660 |
| | *Enterococcus* spp. (n = 2) | 2 | 0.529 | 0 | **0.005** | 2 | **<0.001** | 0 | **0.002** | 2 | **0.001** |

MDR = multi-drug resistant, XDR = extensively-drug resistant, bold numeric = statistically significant (p<0.05).

**Table 6. Statistical evaluation between antimicrobial resistance and biofilm formation.**

| | | MDR | | | XDR | | | Biofilm | | | | | |
| | | | | | | | | MDR | | | XDR | | |
| | | Resistance (%) | | p-value | Resistance (%) | | p-value | Resistance (%) | | p-value | Resistance (%) | | p-value |
| | | Yes | No | | Yes | No | | Yes | No | | Yes | No | |
| Fluoroquinolones | Ciprofloxacin | 63.64 | 69.29 | 0.37 | 95.18 | 50.35 | **<0.001** | 61.04 | 60.32 | 0.931 | 92.31 | 48.51 | **<0.001** |
| | Ofloxacin | 56.57 | 62.99 | 0.327 | 87.95 | 44.06 | **<0.001** | 54.55 | 52.38 | 0.798 | 82.05 | 42.57 | **<0.001** |
| Penicillins | Ampicillin | 98.31 | 86.84 | **0.022** | 95.45 | 93.33 | 0.716 | 100 | 91.67 | 0.55 | 92.86 | 98.11 | 0.304 |
| | Cloxacillin | 30 | 50 | 0.289 | 80 | 27.27 | **0.021** | 26.92 | 66.67 | 0.065 | 80 | 25.93 | **0.019** |
| | Amoxycillin clavulanate | 22.54 | 48.21 | **0.002** | 70.27 | 18.89 | **<0.001** | 18.18 | 45.16 | **0.007** | 73.68 | 14.93 | **<0.001** |
| Cephalorporins | Cefotaxime | 83.84 | 82.4 | 0.776 | 93.83 | 76.92 | **0.001** | 81.82 | 80.33 | 0.824 | 97.3 | 75.25 | **0.003** |
| | Ceftazidime | 86.32 | 86.73 | 0.931 | 98.73 | 79.07 | **<0.001** | 89.19 | 83.64 | 0.356 | 100 | 81.91 | **0.007** |
| | Cefoxitin | 66.67 | 57.14 | 0.635 | 100 | 60.61 | 0.119 | 69.23 | 80 | 0.627 | 100 | 66.67 | 0.17 |
| Aminoglycosides | Gentamicin | 30.3 | 52.8 | **0.001** | 81.48 | 20.98 | **<0.001** | 28.57 | 45.9 | **0.035** | 75.68 | 21.78 | **<0.001** |
| | Amikacin | 40.4 | 51.2 | 0.108 | 71.6 | 32.17 | **<0.001** | 44.16 | 52.46 | 0.332 | 70.27 | 39.6 | **0.001** |
| Others | Imipenem | 29.29 | 56.69 | **<0.001** | 84.34 | 21.68 | **<0.001** | 31.17 | 53.97 | **0.006** | 84.62 | 24.75 | **<0.001** |
| | Cotrimoxazole | 55.79 | 69.91 | **0.035** | 96.2 | 43.41 | **<0.001** | 55.41 | 63.64 | 0.347 | 94.29 | 45.74 | **<0.001** |
| | Chloramphenicol | 32.56 | 53.06 | **0.005** | 76.47 | 24.14 | **<0.001** | 30.77 | 44 | 0.144 | 68.75 | 24.1 | **<0.001** |
| | Erythromycin | 60 | 62.5 | 0.898 | 100 | 54.55 | 0.053 | 61.54 | 83.33 | 0.311 | 100 | 59.26 | 0.078 |
| | Clindamycin | 70 | 50 | 0.343 | 100 | 63.64 | 0.201 | 73.08 | 75 | 0.935 | 100 | 70.37 | 0.271 |

MDR = multi-drug resistance, XDR = extensively-drug resistance, bold numeric = statistically significant (p<0.05).

In this study, 226 bacterial isolates were recovered, with urine (44.69%), sputum (31.42%), and blood (14.6%) samples being the most common sources. Enterobacterales (64.6%) predominated among these isolates, followed by non-fermenters (18.58%) and gram-positive cocci (16.82%). *Citrobacter* spp. (39.04%) was the most prevalent Enterobacterales followed by *E. coli* (35.62%) and *Klebsiella* spp. (16.44%). It should be noted that the majority of these isolates were uropathogens, and their higher incidences among the Enterobacterales could be correlated with higher processing of urine samples. Similar studies have also reported these isolates to be the predominant gram-negative bacteria causing urinary tract infections, septicemia, and other infections [18,19]. The incidences of the ACB complex (11.5%) and *Pseudomonas* spp. (5.3%) among the total isolates in the present study were lower than the results recorded by Dumaru et al [20]. (20% and 12%, respectively). The majority of *Pseudomonas* spp. (62.5%) (p<0.05) were recovered from the sputum samples, which indicates that these groups of organisms are major pathogens causing lower respiratory tract infection (LRTI) [21]. This conclusion correlates well with the results of similar studies conducted in Nepal, which concluded that the majority of isolates in LRTI are gram-negative bacteria, among which *Pseudomonas* spp. account for the maximum number of cases [22,23]. In this study, *S. aureus* (71.05%) represented the most isolated strains of gram-positive cocci, which was inconsistent with the findings of Sultan et al [24]. and Ruchi et al [25]., who reported *Enterococcus* spp. as the commonest gram-positive cocci. The majority of CoNS (88.89) (p<0.05) in this study were isolated from urine samples, which coincides with the findings of Panda et al [10]., who reported *Staphyloccocus* spp. as the predominant bacteria to cause urinary tract infections.

An important global health issue is the emergence of microorganisms that are resistant to antimicrobials, which are used to treat infections. It is estimated that 0.7 million deaths occur annually because of antimicrobial-resistant infections, and this number could rise to 10 million

by 2050, resulting in an approximate economic impact of $100 trillion, primarily affecting LMICs [26]. This study revealed a variable degree of resistance to numerous routinely used drugs. Penicillins (resistance rate: 56.1%) were least effective among the overall bacterial isolates after fluoroquinolones (resistance rate: 63.5%) and cephalosporins (resistance rate: 83.16%), while, aminoglycosides (resistance rate: 44.64%) were moderately effective. Fluoroquinolone resistance among the bacterial isolates could be attributable to mutations either in the DNA gyrase *(gyrA* or *gyrB)*, which have also been identified as the main mechanisms of resistance to fluoroquinolones in all bacterial species, or in the topoisomerase IV, especially in the context of gram-negative bacteria [27]. The higher resistance to cephalosporins (>58.27%) in this study could be ascribed to strains' β-lactam ring, which has a Zwitterionic structure that protects these antibiotics from hydrolysis by the β-lactamases [28]. In this study, both, Enterobacterales (58.43%) showed better sensitivity toward amoxicillin clavulanate, and non-fermenters (54.76%) and gram-positive cocci (63.89%) towards gentamicin. This could be because we use these drugs only to treat infections with MDR bacteria as a last-line therapy in our setting, and because of their infrequent use, the emergence and dissemination of penicillins with beta-lactamase inhibitors- and aminoglycoside-resistant genes are minimal, which restrains the emergence of resistance to these antibiotics.

Among the Enterobacterales, *P. mirabilis* had the highest resistance to ofloxacin (100%) (p<0.05) and cotrimoxazole (100%) (p>0.05). *Citrobacter* spp. (p>0.05), *E. coli* (p>0.05), and *Klebsiella* spp. (p>0.05) showed higher (>80%) resistance rates to ceftazidime, compared to *S.* Typhi (42.86%) (p<0.05). *E. coli* had lower resistance to gentamicin (26.92%) (p<0.05) and amikacin (26.92%) (p<0.05), compared to the resistance rate of *Citrobacter* spp. to gentamicin (59.65%) (p<0.05) and amikacin (63.16%) (p<0.05). The antibiogram of ACB complex in this study showed >90% resistance to cefotaxime (p>0.05) and >70% resistance to imipenem (p<0.05), cotrimoxazole (p>0.05), and ciprofloxacin (p>0.05). Unlike the ACB complex, *P. aeruginosa* showed a higher susceptibility rate to imipenem (75%) (p<0.05), gentamicin (75%) (p<0.05), and ofloxacin (56.25%) (p>0.05). Comparable to our findings, a similar study from Nepal also reported the highest antimicrobial resistance rate against ampicillin (76.7%), ceftazidime (51.5%), cotrimoxazole (48.7%), ciprofloxacin (43.9%), and ofloxacin (41.1%) [29]. Among gram-positive cocci, CoNS showed the highest resistance to ampicillin (100%) (p>0.05) and clindamycin (100%) (p<0.05), while *S. aureus* to ceftazidime (96.3%) (p<0.05) and cotrimoxazole (62.96%) (p<0.05). Another study from Nepal performed among 71 CoNS strains also reported similar resistance rates of the isolates toward penicillin (90%), co-trimoxazole (60%), and azithromycin (60%), and absolute sensitivity to vancomycin and linezolid [30]. Such higher occurrences of antibiotic-resistant isolates in these studies, including ours, could be attributable to the biofilm-forming tendency of the isolates or the dissemination of plasmids carrying antibiotic-resistant genes via the quorum sensing phenomenon [29,31].

In the present study, four *in vitro* phenotypic tests namely, TCPM, TM, CRA, and MCRA were used to detect biofilm formation among the isolated strains. Because these phenotypic tests are less expensive, more time-efficient, easier to perform, and accessible in most laboratory settings, they were chosen over genotypic methods [24]. In the current study, TCPM detected 61.9% of the total bacterial isolates as biofilm producers with 2.65% as strong biofilm producers. In contrast, Panda et al [10]. (11%) and Rampelotto et al [32]. (19.9%) reported a higher incidence of strong biofilm producers in their studies. In this study, the TM detected a higher number of biofilm producers (52.7%) compared to the CRA (38.9%) and the MCRA (38.1%). This study did not correlate strong, moderate, and none/weak biofilm formation in TM, CRA, and MCRA because of their subjective assessment, unlikely to the objective grading scheme used in TCPM [24]. Considering TCPM as the gold standard technique for phenotypic biofilm detection, the TM (p<0.05) showed higher sensitivity (90.8%) and specificity (70.1%)

compared to the sensitivity and specificity of CRA (68.2% and 42%, respectively) (p>0.05) and MCRA (65.1% and 40%, respectively) (p>0.05). The finding of higher sensitivity and specificity of TM compared to CRA and MCRA were also reported in similarly conducted international studies [24,33]. It should be noted, however, that studies conducted by Panda et al. [10] and Hassan et al. [33] demonstrated higher specificity (92.5–95.1%) and PPV (93.3–94.0%) of TM than it did in this study, but lower sensitivity (73–81%) and NPV (66–85.6%). Furthermore, these studies showed higher specificity (>90%) and lower sensitivity (<20%) results for CRA than this study [5,16]. In addition, Panda et al. [10] reported lower sensitivity (17.5%) and higher specificity (94.5%) of MCRA for biofilm detection, which is in contrast to the findings from this study. Such variations in detection parameters in these qualitative tests may be a result of the batch-to-batch variation of the media used as well as subjective errors made during interpretation [24]. In this study, the persistency of black pigmentation in bacterial colonies on CRA, an indication of biofilm formation, declined over time (2–4 days), whereas in MCRA, coloration intensified and remained constant with time (after 3 days). A similar conclusion was reached by Mariana et al. [17] regarding staphylococcal isolates, with the rationale that, by optimizing ingredients and increasing incubation time, the more black pigment was diffused to colonies on CRA. However, our findings do not support the conclusion from Panda et al. [10] stating that MCRA is better at detecting biofilms in only staphylococcal isolates.

In the present study, CoNS (100%) (p>0.05), *Klebsiella* spp. (70.83%) (p>0.05), and the ACB complex (61.54%) (p>0.05) were the highest biofilm producers when detected by TCPM. These findings were supported by the results of Folliero et al. [34], who reported the ACB complex (100%), *K. pneumoniae* (72.7%), and CoNS strains (67.7%) as the major biofilm producers in non-fermenters, Enterobacterales, and gram-positive cocci, respectively. This study showed a higher prevalence of MDR isolates (43.81%) compared to XDR isolates (36.73%). MDR pathogens (55%) accounted for the majority of biofilm producers, with CoNS (88.89%) (p>0.05) being the chief MDR-biofilm producer, followed by *S. aureus* (85.71%) (p>0.05). Half as many XDR pathogens (27.85%) formed biofilms as MDR pathogens did, with *Enterococcus* spp. (100%) (p<0.05) being the most frequent. Meanwhile, Abidi et al. [35]and Amin et al. [36] reported a higher incidence of MDR-biofilm formation in *P. aeruginosa* and the ACB complex, respectively. Considering the existence of a positive correlation between the prevalence of MDR- and XDR-phenotypes among biofilm formations, it would be worthwhile to further investigate the resistance profiles associated with biofilm-producing strains.

In this study, XDR pathogens exhibited higher resistance (>80%) to ciprofloxacin (p<0.001), cefoxitin (p>0.05), gentamicin (p<0.001), imipenem (p<0.001), and cotrimoxazole (p<0.001), while MDR pathogens showed higher resistance (>80%) to ofloxacin (p>0.05), ampicillin (p<0.05), and ceftazidime (p>0.05). Biofilm formers showed higher (>80%) resistance to ampicillin (p>0.05), cefotaxime (p>0.05), and ceftazidime (p>0.05). Similar resistance rates to ampicillin, ciprofloxacin, and cotrimoxazole were found among the biofilm producers in the other studies [28,29]. Data in this study showed a <60% resistance to ofloxacin (p<0.001), cloxacillin (p<0.05), imipenem (p<0.001), and cotrimoxazole (p<0.001) in MDR-biofilm formers, as opposed to XDR-biofilm formers with >80% resistance. These higher antimicrobial resistance rates among MDR- and XDR-biofilm producers may be due to the biofilm matrix's protective function, which reduces antibiotic penetration, or to plasmid-encoded genes being uptaken by bacteria via quorum sensing mechanisms [34,37].

This study suffers from several limitations. First, the biofilm-producing property was not detected among the anaerobes. Second, neither the molecular method nor electron microscopy—two of the most sensitive techniques to identify biofilm formation—were employed to identify biofilm producers. Furthermore, it is significant to note that all of the tests performed

on the bacteria in this study were conducted in vitro, and it is still unclear what will occur in vivo when a biofilm is present. Nevertheless, the infection risks associated with such morpho-types are significant, prolonging hospitalization and increasing treatment failure; therefore, it is necessary to routinely detect bacteria's propensity to form biofilms, and then treat them based on their inhibitory and/or eradication concentrations.

## Conclusion

In this study, a greater number of Enterobacterales were isolated from clinical samples than non-fermenters or gram-positive cocci. Penicillins were the least susceptible antibiotics among the clinical isolates, followed by cephalosporins. Higher incidences of biofilm producers were observed among the clinical isolates. TM demonstrated better sensitivity and specificity results when compared to other phenotypic methodologies. Compared to CRA, the modifications in the agar constituent of MCRA allowed better stability of phenotypic coloration in the bacterial colonies. XDR-biofilm producers were more prevalent than MDR-biofilm producers and had higher rates of antibiotic resistance.

## Author Contributions

**Conceptualization:** Ajaya Basnet.

**Data curation:** Basanta Tamang, Mahendra Raj Shrestha, Pradip Shrestha.

**Formal analysis:** Mahendra Raj Shrestha.

**Investigation:** Ajaya Basnet, Rajendra Maharjan, Sushila Dahal.

**Methodology:** Lok Bahadur Shrestha, Junu Richhinbung Rai.

**Project administration:** Sushila Dahal, Pradip Shrestha.

**Resources:** Basanta Tamang.

**Software:** Sushila Dahal, Shiba Kumar Rai.

**Supervision:** Basanta Tamang, Lok Bahadur Shrestha, Junu Richhinbung Rai, Shiba Kumar Rai.

**Validation:** Ajaya Basnet, Basanta Tamang, Rajendra Maharjan, Shiba Kumar Rai.

**Visualization:** Junu Richhinbung Rai, Rajendra Maharjan, Shiba Kumar Rai.

**Writing – original draft:** Ajaya Basnet.

**Writing – review & editing:** Basanta Tamang, Mahendra Raj Shrestha, Lok Bahadur Shrestha, Junu Richhinbung Rai, Rajendra Maharjan, Pradip Shrestha, Shiba Kumar Rai.

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
