## [Decision Letter · Decision Letter 0]

29 Mar 2023

PONE-D-23-02752Assessment of four in vitro phenotypic biofilm detection methods and antimicrobial resistance in aerobic clinical bacterial isolatesPLOS ONE

Dear Dr. Tamang,

Thank you for submitting your manuscript to PLOS ONE. After careful consideration, we feel that it has merit but does not fully meet PLOS ONE’s publication criteria as it currently stands. Therefore, we invite you to submit a revised version of the manuscript that addresses the points raised during the review process.

We look forward to receiving your revised manuscript.

Kind regards,

Nabi Jomehzadeh, Ph.D (Assistant Professor)

Academic Editor

PLOS ONE

Reviewers' comments:

Reviewer's Responses to Questions

**Comments to the Author**

1. Is the manuscript technically sound, and do the data support the conclusions?

Reviewer #1: Yes

Reviewer #2: Yes

2. Has the statistical analysis been performed appropriately and rigorously? 

Reviewer #1: Yes

Reviewer #2: Yes

3. Have the authors made all data underlying the findings in their manuscript fully available?

Reviewer #1: Yes

Reviewer #2: No

4. Is the manuscript presented in an intelligible fashion and written in standard English?

Reviewer #1: No

Reviewer #2: Yes

5. Review Comments to the Author

Reviewer #1: The author has tried to evaluate the resistance pattern of isolated bacteria in study period and has compared different methods for biofilm formation. Following comments need to be addressed appropriately:

1. Author should clearly mention the aim of the study in abstract and in main manuscript. Presently, the aim defined at two sites is different from each other.

2. In materials and methods, please mention clearly how was the susceptibility testing done for colistin and vancomycin? As per CLSI guidelines, Kirby Bauer method can't be used for these drugs.

3. In results: line 247, the total of samples given is 205, not 194 as mentioned by author.

4. In line 249, Please expand the term ACB because it is written first time here.

5. In table 2; what is the relevance of calculating p-values?

6. Table 6 is not needed. Author has already shown relationship between biofilm formation and drug resistance (MDR & XDR) in table 5.

Reviewer #2: Dear Authors. You are well done.

All the revisions and very important notes are documented on the text. To follow and track and complete your manuscript. It is need to update the references and adding as we mentioned in the text

6. PLOS authors have the option to publish the peer review history of their article (what does this mean?). If published, this will include your full peer review and any attached files.

Reviewer #1: No

Reviewer #2: No

---

## [Author Response · Author response to Decision Letter 0]

15 Apr 2023

General comments (Journal requirements)

Response: We thank you for this useful suggestion. We followed the links and made the necessary changes to the manuscript. As suggested, upon improvements, we have prepared and named three different files as 'Response to Reviewers' (for rebuttal letter), and ' Manuscript with Track Changes’ (for a marked-up copy of the manuscript), and ‘Revised Manuscript’ (for revised paper). Additionally, we have checked and attest that all formatting and style requirements have been met in the manuscript.

2. Please upload a completed version of your questionnaire as Supporting Information when you resubmit your manuscript.

Response: We have uploaded a completed version of questionnaire as “Supporting Information”.

3. Please ensure that you have an ORCID iD and that it is validated in Editorial Manager. 

Response: Validated ORCID iD has been added.

Response: We have now added a full ethics statement in the ‘Methods’ section of the manuscript file, as well as a sentence concerning the written informed consent.

Response to Reviewer 1

1. Author should clearly mention the aim of the study in abstract and in main manuscript. Presently, the aim defined at two sites is different from each other.

Response: First of all, we will like to express our sincere gratitude for your kind suggestions. We appreciate the time and effort that you dedicated to providing feedback on our manuscript and are grateful for the insightful comments on our paper.

Thank you for pointing this out. We agree with this comment. Therefore, following the reviewer suggestions, the aim of the study throughout the manuscript (abstract and main text) has been updated as ... to evaluate diagnostic parameters of four in vitro phenotypic biofilm detection assays in relation to antimicrobial resistance in aerobic clinical bacterial isolates...

2. How was the susceptibility done for colistin and vancomycin? As per CLSI guidelines, Kirby Bauer method can't be used for these drugs.

Response: Thank you for pointing this out. We have now studied the CLSI guideline (32th edition), and as is explicitly stated in it, we have as of this this time amend this error by erasing the susceptibility testing for colistin and vancomycin. Further, we will implement this information in routine diagnosis too.

3. Total is not 194. It is 205. Please confirm.

Response: Dear reviewer, there were 194 samples that tested culture positive. Of these 162 samples yielded monobacterial growth, while 32 samples yielded 2 bacterial growth

162 samples = 162 isolates

32 samples = 64 isolates

194 total samples = 226 total isolates

Therefore, we confirm total samples as 194.

4. Expand ACB.

Response: Thank you for pointing this error. We have expanded the abbreviation accordingly. … Acinetobacter calcoaceticus-baumanii (ACB) complex….

5. What is the relevance of calculating p-value here in this table? 

Response: Thank you for this comment. Actually, the table shows interesting and significant association of pathogen with the samples. For e.g., as we know, E. coli as a significant uropathogen, S. Typhi as a significant bloodstream pathogen, non-fermenters as the significant respiratory pathogens, and lastly S. aureus as the significant wound pathogen; all of these routine findings are explicitly portrayed in the table (significant correlation as per p-value reiterates the preexisting fact). 

6. How do you differentiate between colonizers and pathogens?

Response: As we know, colonization is the presence of bacteria on a body surface (like on the skin, mouth, intestines or airway) without causing disease in the person. Infection is the invasion of a host organism's bodily tissues by disease-causing organisms. The samples obtained in the study were stringently from hospital visiting patients clinically suspected of infection. Hence, they were most likely to be pathogens.

7. Table 6 is not needed. Author has already shown relationship between biofilm formation and drug resistance (MDR & XDR) in table 5.

Response: Dear Reviewer, table 5 is discrete to table 6 in context to simply portraying the pathogen specific incidences of MDR-, XDR- or biofilm formation. The latter distinctly, however, shows antibiotic specific incidences of resistance (%) to each of the tested, be it with regard to MDR-, XDR-, or biofilm-MDR-, or biofilm-XDR-isolates. Furthermore, table 6 summarizes the overall resistance to each antibiotic tested, while table 5 fundamentally shows the frequencies. Therefore, we sincerely request you to reconsider the urgency of table 6.

Response to Reviewer 2

1. All the revisions and very important notes are documented on the text. To follow and track and complete your manuscript. It is need to update the references and adding as we mentioned in the text

Thank you for your kind suggestions. We have now updated the title as “Assessment of four in vitro phenotypic biofilm detection methods in relation to antimicrobial resistance in aerobic clinical bacterial isolates” We have added all the mentioned references in the study as well as the indicated change(s).

---

## [Decision Letter · Decision Letter 1]

13 Oct 2023

PONE-D-23-02752R1Assessment of four in vitro phenotypic biofilm detection methods in relation to antimicrobial resistance in aerobic clinical bacterial isolatesPLOS ONE

Dear Dr. Tamang,

Thank you for submitting your manuscript to PLOS ONE. After careful consideration, we feel that it has merit but does not fully meet PLOS ONE’s publication criteria as it currently stands. Therefore, we invite you to submit a revised version of the manuscript that addresses the points raised during the review process.

We look forward to receiving your revised manuscript.

Kind regards,

Nabi Jomehzadeh, Ph.D (Assistant Professor)

Academic Editor

PLOS ONE

Journal Requirements:

Additional Editor Comments (if provided):

"We note that one or more reviewers has recommended that you cite specific previously published works. As always, we recommend that you please review and evaluate the requested works to determine whether they are relevant and should be cited. It is not a requirement to cite these works. We appreciate your attention to this request.”

Reviewers' comments:

Reviewer's Responses to Questions

**Comments to the Author**

1. If the authors have adequately addressed your comments raised in a previous round of review and you feel that this manuscript is now acceptable for publication, you may indicate that here to bypass the “Comments to the Author” section, enter your conflict of interest statement in the “Confidential to Editor” section, and submit your "Accept" recommendation.

Reviewer #1: All comments have been addressed

Reviewer #2: All comments have been addressed

Reviewer #3: All comments have been addressed

2. Is the manuscript technically sound, and do the data support the conclusions?

Reviewer #1: Yes

Reviewer #2: Yes

Reviewer #3: Yes

3. Has the statistical analysis been performed appropriately and rigorously? 

Reviewer #1: Yes

Reviewer #2: Yes

Reviewer #3: Yes

4. Have the authors made all data underlying the findings in their manuscript fully available?

Reviewer #1: Yes

Reviewer #2: Yes

Reviewer #3: Yes

5. Is the manuscript presented in an intelligible fashion and written in standard English?

Reviewer #1: Yes

Reviewer #2: Yes

Reviewer #3: Yes

6. Review Comments to the Author

Reviewer #1: the author has addressed all the comments and has made appropriate changes. The author has provided the data in well-understandable manner and has included the tables appropriately.

Reviewer #2: Dear Author

You are doing well

There are simple errors in writing the two references in numbers 13 and 41 in the list of references of the revised manuscript which the contributing authors are missing. Thus, the below the revisions:-

1-Reference 13:- The corrected and revised form of reference 13 is as follow:-Al-Ouqaili, M.T.S., Al-Kubaisy, S.H.M., Al-Ani, N.F.I. Biofilm antimicrobial susceptibility pattern for selected antimicrobial agents against planktonic and sessile cells of clinical isolates of staphylococci using MICs, BICs, and MBECs. Asian Journal of Pharmaceutics, Volume 12, Issue 4, October-December 2018, Pages S1375-S1383 instead of the citation form in the list of references.

2-Reference 14:- The corrected and revised form of reference 41 is as follow:-Al-Ouqaili, MTS, Al-Taei, SA, Al-Najjar A. Molecular Detection of Medically Important Carbapenemases Genes Expressed by Metallo-β-lactamase Producer Isolates of Pseudomonas aeruginosa and Klebsiella pneumoniae. Asian Journal of Pharmaceutics • Jul -Sep 2018 (Suppl ) • 12 (3) | S991 instead of the citation form in the list of references.

Reviewer #3: The authors have well addressed the reviewer’s comments however there are minor clarifications in results required from authors.

Table 2: why no of isolates are more than no of samples of urine blood and sputum

Table 2: what is the significance of p-value

Table 3: what is the significance of bold figures. e.g., in S. Typhi ceftazidine in bold with 3 resistant and not ciprofloxacin etc with 6 resistant isolates. Similar is the case with S. aureus and P. aeruginosa.

7. PLOS authors have the option to publish the peer review history of their article (what does this mean?). If published, this will include your full peer review and any attached files.

Reviewer #1: No

Reviewer #2: No

Reviewer #3: **Yes: **Abu Baker Siddique

---

## [Author Response · Author response to Decision Letter 1]

14 Oct 2023

We are grateful to the reviewers for their insightful comments on this study. We have been able to incorporate changes to reflect most of the suggestions provided by the reviewers.

---

## [Decision Letter · Decision Letter 2]

7 Nov 2023

Assessment of four in vitro phenotypic biofilm detection methods in relation to antimicrobial resistance in aerobic clinical bacterial isolates

PONE-D-23-02752R2

Dear Dr. Tamang,

We’re pleased to inform you that your manuscript has been judged scientifically suitable for publication and will be formally accepted for publication once it meets all outstanding technical requirements.

Kind regards,

Nabi Jomehzadeh, Ph.D (Assistant Professor)

Academic Editor

PLOS ONE

Additional Editor Comments (optional):

Reviewers' comments:

Reviewer's Responses to Questions

**Comments to the Author**

1. If the authors have adequately addressed your comments raised in a previous round of review and you feel that this manuscript is now acceptable for publication, you may indicate that here to bypass the “Comments to the Author” section, enter your conflict of interest statement in the “Confidential to Editor” section, and submit your "Accept" recommendation.

Reviewer #3: All comments have been addressed

2. Is the manuscript technically sound, and do the data support the conclusions?

Reviewer #3: Yes

3. Has the statistical analysis been performed appropriately and rigorously? 

Reviewer #3: Yes

4. Have the authors made all data underlying the findings in their manuscript fully available?

Reviewer #3: Yes

5. Is the manuscript presented in an intelligible fashion and written in standard English?

Reviewer #3: Yes

6. Review Comments to the Author

Reviewer #3: (No Response)

7. PLOS authors have the option to publish the peer review history of their article (what does this mean?). If published, this will include your full peer review and any attached files.

Reviewer #3: No

---

## [Editor Report · Acceptance letter]

13 Nov 2023

PONE-D-23-02752R2 

Assessment of four *in vitro* phenotypic biofilm detection methods in relation to antimicrobial resistance in aerobic clinical bacterial isolates 

Dear Dr. Tamang:

I'm pleased to inform you that your manuscript has been deemed suitable for publication in PLOS ONE. Congratulations! Your manuscript is now with our production department. 

Kind regards, 

on behalf of

Dr. Nabi Jomehzadeh 

Academic Editor

PLOS ONE